# The Influence of Taste Genes on Body Fat and Alcohol Consumption

**DOI:** 10.3390/nu16111756

**Published:** 2024-06-04

**Authors:** Mohammad K. Shushari, Tianlan Wei, Pradtana Tapanee, Diane Tidwell, Terezie Tolar-Peterson

**Affiliations:** 1Department of Food Science, Nutrition, and Health Promotion, Mississippi State University, Starkville, MS 39762, USA; mks543@msstate.edu (M.K.S.); dtidwell@fsnhp.msstate.edu (D.T.); 2Department of Counseling, Educational Psychology, and Foundations, Mississippi State University, Starkville, MS 39759, USA; tw1518@msstate.edu; 3Institute of Nutrition, Mahidol University, Nakhon Pathom 73170, Thailand; pradtana.tap@mahidol.edu; 4Department of Health Science and Human Ecology, California State University San Bernardino, San Bernardino, CA 92407, USA

**Keywords:** alcohol consumption, college students, taste genes, obesity, ethnicity, gender

## Abstract

Dietary intake and alcohol consumption might be influenced by genetic variations in taste receptor genes. The objectives of this study were to examine the relationship between polymorphisms in the bitter taste receptor genes TAS2R13 (rs1015443) and TAS2R38 (rs1726866, rs10246939, and rs713598) as well as alcohol consumption and body fat percentage in college students. Four hundred and two students with a mean age of 20.2 years participated in this study. An NIH Diet History Questionnaire (DHQ II) was used to collect data on their dietary intake, while an AUDIT survey was used to determine their level of alcohol consumption. Bitter taste receptor gene polymorphisms were assessed by TaqMan allelic discrimination assays. Despite significant associations between TAS2R13 (rs1015443) and certain aspects of alcohol consumption, including the frequency of alcohol intake, no significant associations were found between TAS2R13 (rs1015443) and alcohol consumption after accounting for confounding variables in the regression model. Neither association was found regarding percent of body fat. In contrast, ethnicity and gender significantly influenced percent of body fat (*p* < 0.001), while no significant association was observed between TAS2R13 (rs1015443) and percent of body fat. Likewise, TAS2R38 (rs1726866, rs10246939, and rs713598) demonstrated no significant association with alcohol consumption and percent of body fat. These results were controlled for confounding factors, such as ethnicity and gender. Body fat percentage and alcohol consumption may be influenced by ethnicity, gender, and age rather than SNPs of TAS2R13 and TAS2R38 genes. Assessing taste genes’ interactions with diet and body composition might be useful in identifying human disease risk.

## 1. Introduction

People perceive food primarily based on sight, odor, taste, and sound, and, of these, taste has the greatest influence on food intake [1]. Humans can sense five established basic tastes, including sweet, sour, salty, bitter, and umami, all of which develop in childhood and continue to evolve throughout our lifespan, influencing food preference [2]. Moreover, taste perception not only influences the quality of food intake, but also the quantity [3], both of which affect our health and directly contribute to nutrient-related health outcomes. Bitter taste is thought to have evolved to detect toxic compounds and is considered one of the most sensitive tastes [4]. Bitter compounds are perceived through the TAS2R family of taste receptors, and polymorphisms in these taste receptors might influence how humans perceive bitter taste. A previous study confirmed that the variation in the TAS2R38 gene mediates the bitter taste of thiourea compounds, such as phenylthiocarbamide (PTC) and 6-n-propylthiouracil (PROP) [5]. People who are supertasters can perceive intense bitter tastes from concentrated PTC or PROP, while non-tasters will detect little to nothing. In addition, the TAS2R13 gene, a member of the G-protein coupled receptor taste superfamily, also corresponds to bitter taste [6]. A study published in 2012 revealed the association between TAS2R13 (rs1015443 [C1040T, Ser259Asn]) and alcohol consumption, measured via the Alcohol Use Disorders Identification Test (AUDIT), in patients with head and neck cancer [7]. 

As taste perceptions are related to food intake, the balance of food intake is closely linked to body fat. According to Bouthoorn et al. (2014), TAS2R38 (rs713598) mediated an association between body fat percentage and PROP status in six-year-old girls. The researchers further investigated whether there is an association between body composition (fat mass and BMI) and PROP taste ability using a prospective cohort design. The study found that girls who were non-tasters had a higher body fat percentage and body weight than their supertaster counterparts. On the contrary, boys’ body weights were not associated with PROP status. In another study conducted with children, Keller and Tepper [8] discovered an association between bitter taste and food consumption. In the study, the percentile of weight per height was higher for non-taster boys who reportedly consumed more protein and fat [8]. Tepper and colleagues, in yet another study, reported that young non-taster women, identified by the PROP taste test, consumed more energy but not fat from a food buffet [9]. The higher consumption of energy-dense foods was associated with the increased adiposity of non-tasters compared to taster women. 

It is important to acknowledge that the selection of TAS2R13 and TAS2R38 was based on prior research that indicated significant associations between these genes and dietary behaviors, these two genes being among numerous bitter receptor taste genes. Thus, further research should consider the broad span of functional TAS2R genes and the potential of sweet taste genes’ influences on taste perception. This study focused on a college student population characterized by diverse socioeconomic backgrounds, healthcare patterns, and lifestyles compared to previous study populations (i.e., patients or children). 

The primary aim of this study was to elucidate the connections among various variables: TAS2R13 and TAS2R38 haplotype distribution, alcohol consumption, and body fat percentage. In particular, this research sought to determine the frequency distribution of TAS2R13 (rs1015443) among college students, examine the association between polymorphisms in TAS2R13 (rs1015443) and TAS2R38 (rs1726866, rs10246939, and rs713598), and assess their potential impact on alcohol consumption and body fat percentage. Therefore, the purpose of this comprehensive analysis was to clarify the genetic influence that may underlie dietary behaviors and health outcomes. 

## 2. Materials and Methods

### 2.1. Overview

This study is part of the ongoing research project “BODY AP: Biological factors for Obesity Development in Young Adults Project”, which is focused on the association between biological, environmental, and socioeconomic factors as well as the body composition of young adults. One of the project’s aims was to determine whether the bitter taste receptor genes TAS2R13 (rs1015443) and TAS2R38 (rs1726866, rs10246939, and rs713598) are associated with alcohol intake and body fat percentage. The reason for choosing these genes, alcohol, and body fat percentage was that previous research showed significant associations between them. The study recognizes the limitations of not including other bitter or sweet taste receptors, which may also contribute to variations in taste perception and dietary behaviors. The participants were students attending Mississippi State University in the United States, aged 18–42 years. All participants were required to make a one-time visit to the campus-based laboratory to complete body composition measurements, provide a saliva sample, and complete related surveys. Data were collected between February 2016 and November 2020. The study protocol was approved by the Ethics Committee of the university pertaining to conducting research with human participants. 

### 2.2. Alcohol Intake and Behaviors

#### 2.2.1. Diet History Questionnaire

Participants were asked to complete the food frequency questionnaire NIH Web-DHQ-II, which includes 153 items [10]. The questionnaire tool (DHQ) gathered data that described participants’ food intake and portion sizes over the past 12 months and were evaluated using Nutrient Database and Diet * Calc software version 1.5.0. 

#### 2.2.2. AUDIT

The 10-item AUDIT screening tool, developed by WHO, was used to collect data from participants on drinking behaviors, alcohol consumption, and pertinent alcohol-related problems [11]. In this study, the research team extracted the first three questions from a total of 10 questions. This is called the AUDIT-c, which is a brief and effective tool for evaluating alcohol consumption versus the classic 10-question questionnaire [12]. Participants’ responses to those three alcohol consumption questions (q1: “How often do you have a drink containing alcohol?”; q2: “How many drinks containing alcohol do you have on a typical day?”; and q3: “How often do you have five or more drinks on one occasion?”) were analyzed. For the AUDIT-c questionnaire tool, sum scores can range from 0 to 12, which quantifies alcohol intake [13]. The type of alcohol intake included beer, spirits, and wine. The typical serving, frequency, and quantity were recorded; then, alcohol intake was calculated by converting the reported quantity into grams/day [10]. 

### 2.3. Body Composition Testing

Bioelectrical impedance analysis (MC-780, Tanita Corporation, Tokyo Japan) was used to estimate body weight, total body fat percentage, and fat-free mass. Body fat percent was obtained based on the relationship between fat content and body composition. Impedance (Z) measures the electric impulse resistance when passing through tissues across the feet, legs, and abdomen. The measures were applied to validated Tanita equations, considering inductance and capacitance. 

### 2.4. Genetic Analysis 

Saliva was collected using the Salimetrics system (Salimetrics, State College, PA, USA). Each participant was asked to provide two saliva samples in 2 mL cryovials. Saliva was blotted onto P5 filter paper (Fisher brand, Seattle, WA, USA) and allowed to dry for subsequent DNA extraction. DNA was extracted using the DNA Extract All Reagents Kit (Applied Biosystems, Foster City, CA, USA). Genotyping was conducted using TaqMan allelic discrimination assays and the QuantStudio5 real-time PCR system (Applied Biosystems, Foster City, CA, USA).

### 2.5. Statistical Analysis

All data were entered in SPSS (version 27, IBM, Armonk, NY, USA), while missing data were excluded from the final analysis. Descriptive statistics were computed, and the alpha level was set at 0.05 for all inferential statistics. Chi-square goodness-of-fit tests were conducted to analyze major and minor allele frequencies and compare them to the general US population. Two-tailed independent *t*-tests were conducted to check if there were any differences in alcohol consumption between students of legal age (≥21 years) and underage (<21 years) students. Spearman’s rho was computed to measure the association between AUDIT-c and fat percent. Multiple linear regression was carried out to execute a model of the influence of ethnicity, age, and SNP rs1015443 on the sum score of AUDIT-c and grams of alcohol consumption per day. Two-way between-subjects ANOVA tests were conducted to explain the effect of bitter taste SNPs and ethnicity on body fat percentage. For the ANOVAs, Levene’s test determined the assumption of homogeneity, and Tukey HSD adjustment was applied in the post hoc tests. Differences in body fat percentage by gender were tested through an independent-samples *t*-test. For further testing on the influence of the independent variables (SNP rs1015443, ethnicity, and gender) on the dependent variable (body fat percent), the research team conducted a multiple linear regression analysis, controlling for gender and ethnicity. 

## 3. Results 

### 3.1. Participant Characteristics 

This study included 422 participants who self-reported that they were healthy, 7 of whom were excluded due to missing ethnicity information, while another 13 were removed due to reporting an ethnicity other than Caucasian or African American. Consequently, a total of 402 participants of two ethnicities (297 Caucasians and 105 African Americans) were retained for our final analysis (Table 1). 

### 3.2. Allelic Distribution among the Participants

The major and minor allele frequency distribution for all participants is presented in Table 2 with comparisons to the American population. Minor and major frequency alleles were calculated based on Hardy Weinberg equations: TT = 0.48, CC = 0.52, alpha level = 0.05, critical value = 5.99. 

### 3.3. Alcohol Consumption per Age Stratification

To determine whether alcohol consumption, measured by AUDIT questions, differed between students of legal drinking age (≥21 years) and underage students (18–20 years), two-tailed independent-samples *t*-tests were conducted. Responses to question 1 (q1) of the AUDIT-c (“How often do you have a drink containing alcohol?”) were significantly different (t [400] = −4.354, *p* < 0.001) between students ≥ 21 years (mean [μ ± SD] intake of 0.88 ± 1.023 drinks) and students aged 18–20 years (mean intake of 2.43 ± 0.958 drinks). There was no significant difference in responses to q2 (“How many drinks containing alcohol do you have on a typical day when you are drinking?”) between students aged ≥ 21 years (1.71 ± 0.911 drinks) and students aged 18–20 years (1.69 ± 0.823). Likewise, responses to q3 (“How often do you have five or more drinks on one occasion?”) were not significantly different between students aged ≥ 21 years (1.84 ± 0.903) and students aged 18–20 years (1.71 ± 0.857).

Data from the DHQ II revealed that there was also no significant difference in grams of alcohol consumed per day (8.10 ± 15.04) between students aged 18–20 years and (12.5 ± 32.3) students ≥ 21 years. Also, there was no significant difference in the percentage of energy intake from alcohol per day (1820.2 ± 1326.6 kcal for students aged 18–20 years and (2040.1 ± 1699.4) kcal for students aged ≥ 21 years). Due to the significant difference observed in the scores for q1 of the AUDIT-c between the age groups, the sample was stratified into age groups for all subsequent analyses: Group 1 (age 18–20 years) and Group 2 (age ≥ 21 years).

### 3.4. Testing the SNPs as a Function of AUDIT-c, Alcohol Consumption, Energy, and Fat Percentage

TAS2R SNPs were analyzed using the Kruskal–Wallis test for any association with AUDIT-c responses, alcohol consumption as measured by the DHQ II, and body fat percentage as measured by TANITA. TAS2R38 SNPs (rs1726866, rs10246939, and rs713598) were not significantly associated with AUDIT-c or DHQ II measures (the association remained non-significant even when Caucasian ethnicity alone was considered). However, TAS2R13 (rs1015443) was significantly associated with q2 (“How many drinks containing alcohol do you have on a typical day?”), q3 (“How often do you have five or more drinks on one occasion?”), and q1 (“How often do you have a drink containing alcohol?”) of the AUDIT-c. TAS2R13 haplotype distributions and their association with alcohol consumption are explained in Table 3.

Using two-way ANOVA for TAS2R13 (rs1015443) and ethnicity for the effects on alcohol consumption, the results showed that q1 (“frequency of consumption”) had a small effect size (F (5,396) = 4.338, *p* < 0.001) for the overall model and an even smaller effect size (F (2,396) = 0.553, *p* = 0.576) for the interaction effect. Ethnicity, which included Caucasians and African Americans, had F (1,396) = 9.532, *p* = 0.002, while TAS2R13 (rs1015443) had F (2,396) = 0.573, *p* = 0.564. Allelic distribution between the ethnicities was generated as follows: TT—Caucasian (55) and African American (65); CT—Caucasian (141) and African American (35); CC—Caucasian (101) and African American (5). Question 2 (“consumption on a typical day”) had the same level of significance with the following allelic distribution: TT—Caucasian (55) and African American (65); CT—Caucasian (141) and African American (35); CC—Caucasian (101) and African American (5). Question 3 (“more than five drinks”) had the following allelic distribution: TT—Caucasian (55) and African American (65); CT—Caucasian (141) and African American (35); CC—Caucasian (101) and African American (5).

For the DHQ II questionnaire, TAS2R13 (rs1015443) was significantly associated with the percentage of energy intake from alcohol (*p* = 0.012) and alcohol consumption per gram (*p* = 0.027). To assess the relationship between age and the sum of AUDIT-c, simple linear regression was applied. The assumptions of linearity between variables, independent observations, and homoscedasticity were met. There was no significant association between age and the sum of AUDIT-c (r = 9.6%, *p* = 0.055). To better explain the effect of ethnicity, age, and TAS2R13 (rs1015443) on alcohol consumption per gram (measured using DHQ II), the researchers conducted hierarchical linear regression (age and ethnicity as blocking factors) and excluded missing observations using the “listwise exclude” function. The R square was 3.1% for the blocking factors and 3.2% for both the predictive variables and the variables that were controlled. After controlling for ethnicity and age, the variability of alcohol consumed per gram was explained by SNP rs1015443 (*p* = 0.001), indicating little effect of SNP rs1015443 (F (3) = 4.306). ANOVA determined that the overall model can be predictive of alcohol consumption (*p* = 0.005).

TAS2R38 (rs1726866, rs10246939, and rs713598) and TAS2R13 (rs1015443) were not significantly associated with body fat percentage (*p* = 0.252). In African Americans, mean body fat (mean ± SD) was 30.39 ± 11.64, 27.62 ± 11.08, and 28.46 ± 9.91 for TT, CT, and CC allelic genotypes of TAS2R13 (rs1015443), respectively. Meanwhile, in Caucasians, it was 27.28 ± 7.46, 24.99 ± 8.26, and 26.34 ± 9.15 for TT, CT, and CC alleles, respectively.

## 4. Discussion

The study’s objectives were to determine TAS2R13 (rs1015443) frequency distributions among college students and examine the association between alcohol consumption and TAS2R13 (rs1015443) and TAS2R38 (rs1726866, rs10246939, and rs713598), as well as investigate the effect of TAS2R13 and TAS2R38 polymorphisms on body fat percentage. It is imperative to consider the broader spectrum of unexplored sweet taste genes and other bitter taste receptors, which could enhance our understanding of more comprehensive genetic factors and associations with dietary behavior.

This study recruited 422 healthy participants but retained 402 for final analysis after excluding 7 participants due to missing the ethnicity variable and another 13 due to their being of ethnicities other than Caucasian or African American. Women were overrepresented, constituting 84.6% of the participants. Otherwise, the study sample was comparable to the general US population in major and minor allele frequencies.

Major and minor allele frequencies in the different ethnic groups were analyzed, revealing no significant difference between the allele frequency distribution in the study population (encompassing both ethnic groups) and that in the general population (χ^2^ = 0.016, *p* = 0.99 [<5.99 critical value]) (Table 2). The difference between TT (more frequent among African Americans) and CT (more frequent among Caucasians) was significant (*p* = 0.013).

The associations between TAS2R SNPs as well as alcohol consumption (AUDIT-c) and energy from alcohol (DHQ II) were examined. The TAS2R38 gene (rs1726866, rs10246939, and rs713598) demonstrated no significant association with alcohol consumption. However, the gene TAS2R13 (rs1015443) showed a significant association with AUDIT-c for the number of drinks consumed on a typical day (*p* = 0.050), instances of consuming ≥ 5 drinks at a time (*p* = 0.031), and the frequency of consumption over the day (*p* = 0.041). These findings are consistent with the study by Duffy and Hayes (2010), which reported that genetic variations related to bitter taste were associated with alcohol intake. The DHQ II (grams of alcohol and energy consumed from alcohol), and the allelic distribution of TAS2R13 (rs1015443) were significantly associated (*p* = 0.027) (Caucasian: TT 13.23 ± 46.68, CT 10.21 ± 16.15, CC 10.27 ± 16.80; African American: TT 4.74 ± 10.92, CT 8.91 ± 15.05, CC 2.90 ± 3.36). The energy from alcohol was significantly associated with this SNP (*p* = 0.012). Dotson et al. (2012) [7] published a study on patients with head and neck cancer and reported that rs1015443 was associated with alcohol consumption. However, they did not generalize their findings because of the potential influence of radiotherapy on taste palatability. Allen et al. [14] also found that ethanol mouth taste intensity is related to TAS2R13 (rs1015443). In this study, the results of the regression analysis indicated the predictors explained 3.2% of the variance (R^2^ = 0.032, F (3,397) = 4.306, *p* = 0.005). It was observed that age (β = 1.498, *p* = 0.003) significantly predicted alcohol consumption per gram, while the effect of TAS2R13 rs1015443 was not significant (*p* = 0.676). When changing the dependent variable to sum AUDIT-c, the significance remained the same, but the coefficient of variation became −0.231 for ethnicity, considering 1 as the base level (Caucasian). However, holding all other variables constant, being an African American was observed to decrease alcohol consumption by 0.231 × 3. Therefore, rs1015443 was not significantly associated with alcohol consumption, which, although it contradicts the findings of Dotson et al. [7], corroborates their explanation that radiotherapy might have interfered with taste palatability.

Conversely, the two-way ANOVA showed a significant effect of TAS2R13 (rs1015443) and ethnicity on percent of body fat (*p* = 0.008) as a model, although the independent effect of the variables was not significant; navigating through the multiple comparisons among means, there was a significant difference between the two alleles (TT) and (CT) (*p* = 0.005). Notably, the homogeneity assumption was not met (through the residual plot analysis), but there was no multicollinearity issue in the model (VIF < 10) and the q-q-plot was normal. As expected, gender was significantly associated with the percent of body fat (*p* = 0.001). Then, we applied multiple linear regression while controlling for the confounding variables of gender and ethnicity. As a result, the association of body fat percentage with gender and ethnicity remained significant (*p* < 0.001), but not that with TAS2R13 (rs1015443) (*p* = 0.802). Some of the assumptions were violated, and applying transformational methods did not fix the predictive model. Therefore, these findings do not suggest an association between the bitter taste gene TAS2R13 (rs1015443) and body fat percentage.

Also, this study could not support the findings presented by Bouthoorn et al. [15] on the association between TAS2R38 (rs713598), *p* = 0.760, and body fat percentage. The other SNPs (rs1726866, *p* = 0.068, and rs10246939, *p* = 0.745) were also not significant. Unlike the study of Bouthoorn et al. [15], which was based on children, this study design considered adult participants and did not assess bitter taste phenotypes, which may suggest a diminished influence of TAS2R38 (rs713598) on body fat percentage after passing through childhood.

A limitation of the study is that the sample included mostly female participants. We also did not include a test of bitter taste perception. However, the sample size was relatively large and included both genders and two different ethnicities. In addition, the data collection combined both subjective and objective measures through self-reported questionnaires, measurement of body composition, and genotyping the DNA of human participants.

## 5. Conclusions

Findings related to TAS2R bitter taste genes might be generalized to the healthy population. Contrary to previously published research (Bouthoorn et al. [15]), TAS2R38 SNPs were not significantly associated with body fat percentage, nor did we observe a consistent association between TAS2R13 (rs1015443) and alcohol consumption after controlling for demographic factors, in contrast with the study of Dotson et al. [7]. There was a significant difference in TAS2R13 (rs1015443) allele frequencies between ethnicities but not between genders. TAS2R13 (rs1015443) did not demonstrate a significant association with alcohol consumption when the ≥21 age group was considered, and the overall regression model clarified that the significant association observed earlier resulted from demographic factors. There was no difference in the TAS2R13 (rs1015443) allele frequency distributions among different ethnicities in the general population and in our research population. It is worth noting that the lack of significant associations does not imply any causation or absolute absence of association; therefore, a larger study with more objective measures, such as actual taste tests and a wider array of both bitter and sweet taste receptor genes, is recommended.

## Figures and Tables

**Table 1 nutrients-16-01756-t001:** Participants’ demographics.

Variables	N (%)
Gender	
Male	62 (15.4%)
Female	340 (84.6%)
Race	
Caucasian	297 (73.9%)
African American	105 (26.1%)
Age	20.2 ± 2.23 ^a^
Height ^b^	5.51 ± 0.28
Weight ^c^	149.5 ± 38.7
BMI	24.3 ± 5.70
Fat mass ^c^	42.6 ± 27.1
FFM ^d^	109.3 ± 53.6
Alcohol consumption ^e^	9.55 ± 22.3

^a^ Mean ± SEM (standard error of the mean). ^b^ Height in feet/inches. ^c^ Weight and fat mass in pounds. ^d^ FFT: fat-free mass in pounds. ^e^ Grams/day.

**Table 2 nutrients-16-01756-t002:** *TAS2R13* (rs1015443) and TAS2R38 (rs1726866, rs10246939, rs713598) allele frequencies compared to the American population, N = 402.

Ethnicity	rs1015443 Genotype	N (%)	Allele Frequency	χ^2^	*p* Value
Caucasian	TT	55	0.42	0.219	0.896
	CT	141			
	CC	101	0.58		
African American	TT	55	0.76	0.034	0.982
	CT	35			
	CC	5	0.24		
TAS2R38 rs1726866					
Caucasian	AA	81	0.512	0.0041	0.948
	AG	142			
	GG	74	0.488		
African American	AA	14	0.371	0.0476	0.828
	AG	50			
	GG	41	0.629		
rs10246939					
Caucasian	CC	74	0.485	0.0001	0.992
	CT	140			
	TT	83	0.515		
African American	CC	15	0.405	0.0225	0.881
	CT	55			
	TT	35	0.595		
rs713598					
Caucasian	CC	100	0.574	0.0355	0.850
	CG	141			
	GG	56	0.426		
African American	CC	35	0.595	0.0529	0.818
	CG	55			
	GG	15	0.405		

**Table 3 nutrients-16-01756-t003:** (**A**) The effect of *TAS2R13* (rs1015443) genotype on AUDIT-c responses, energy intake from alcohol, and alcohol consumption of students of different age groups (*n* = 402). (**B**) The effect of *TAS2R38* genotype on AUDIT-c responses, energy intake from alcohol, and alcohol consumption of students of different age groups (*n* = 402).

(A)			
Measures	Age Categories		
	All	18–20 Years	≥21
Q1: “How often do you have a drink containing alcohol?”	*p* value = 0.041 Pairwise test comparison Bonferroni correction TT-CT: −2.077 TT-CC: −2.315 CT-CC: −0.510 (non-significant)	*p* value = 0.199	*p* value = 0.077
Q2: “How many drinks containing alcohol do you have on a typical day when drinking?”	*p* value = 0.058	*p* value = 0.021	*p* value = 0.824
Q3: “How often do you have five or more drinks on one occasion?”	*p* value = 0.031 Pairwise test comparison TT-CT: −1.488 TT-CC: −2.638 CT-CC: 0.926 (non-significant)	*p* value = 0.023	*p* value = 0.644
Alcohol consumption per gram USDA	*p* value = 0.035 Pairwise test comparison TT-CT: −2.332 TT-CC: −2.182 CT-CC: −0.117 (non-significant)	*p* value = 0.045 Mean rank TT 116.2 CT 141.7 CC 142.7	*p* value = 0.361 TT 60.5 CT 69.7 CC 72.2
Energy from alcohol (grams)	*p* value = 0.012 Pairwise test comparison TT-CT: −2.440 TT-CC: −2.720 CT-CC: −0.597 (non-significant)	*p* value = 0.029 Mean rank TT 115.4 CT 140.4 CC 145.7	*p* value = 0.192 TT 58.7 CT 70.2 CC 73.7
** (B) **			
	**rs1726866**	** rs10246939 **	** rs713598 **
	***p* Value**		
Q1: “How often do you have a drink containing alcohol?”	0.194	0.605	0.904
Q2: “How many drinks containing alcohol do you have on a typical day when drinking?”	0.175	0.218	0.445
Q3: “How often do you have five or more drinks on one occasion?”	0.435	0.991	0.686
Alcohol consumption per grams/day USDA	0.692	0.984	0.766
Energy from alcohol (grams/day)	0.480	0.998	0.706

*p* value, pairwise comparison, and mean rank per gram.

## Data Availability

The original contributions presented in the study are included in the article, further inquiries can be directed to the corresponding author.

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
