# Peer review of "The Influence of Taste Genes on Body Fat and Alcohol Consumption"

_nutrients, 2024, doi:10.3390/nu16111756_

Round 1

Reviewer 1 Report

Comments and Suggestions for Authors

Influence of Taste Genes:  Shushari et al  May 8, 2024-Nutrients

Abstract:

Line 14: Should TAS2R13 (rs101544)-significant association- be listed in your list of polymorphisms?

Line 21: delete word “indeed”

Introduction: Seems ok, but could better specifically discuss genes of interest in your college-aged study population.

Line 37: I think you mean to say we develop our taste for these five taste qualities, but that our preferences continue to evolve over our lifespan correct?

Purpose statement at end of Introduction could be made more specific ….this part seemed a bit vague”… probes the objectives of 67 this study to determine TAS2R13…”  You covered this much better in first statement of conclusions however: “The study objectives were to determine TAS2R13 (rs1015443) frequency distributions among college students, examine the association between alcohol consumption and TAS2R13 (rs1015443) and TAS2R38 (rs1726866, rs10246939, and rs713598), as well as investigate the effect of TAS2R13 and TAS2R38 polymorphisms on body fat percentage.

Methods: Seems pretty well written and statistical analysis portion very thorough.

Line 80: Authors may wish to just say the state where the data was obtained, not everyone will agree what “mid-southern” really is.  In the acknowledgements you describe Mississippi State University, may as well say this in methods as well.

Line 157-159: Perhaps I missed this item, did your stusy data set include persons who did not drink alcohol or perhaps once/month?  A range of alcohol intake with mean st dev would be helpful…

Conclusions:  May wish to suggest that future studies include an actual taste test to examine correlations between TAS2R and ones actual ability to perceive this taste quality.

Comments on the Quality of English Language

Writing is fine

Author Response

Thank you very much for taking the time to edit our work. The recommendations were addressed as follows:

Abstract:

Line 14: TAS2R13 (rs1015443) has been included in the list of polymorphisms as a significant association.

Line 21: The word “indeed” has been deleted.

Introduction:

Lines 70-72 have been updated to better discuss the genes of interest in our college-aged study population.

Line 37: The statement has been corrected to indicate that we develop our taste for these five taste qualities, but our preferences continue to evolve over our lifespan.

The purpose statement at the end of the Introduction has been made more specific, reflecting the objectives to determine TAS2R13 (rs1015443) frequency distributions among college students, examine the association between alcohol consumption and TAS2R13 (rs1015443) and TAS2R38 (rs1726866, rs10246939, and rs713598), and investigate the effect of TAS2R13 and TAS2R38 polymorphisms on body fat percentage. This has been clarified similarly to the first statement of conclusions.

Methods:

Line 80: The state where the data was obtained is now specified as Mississippi, aligning with the acknowledgments which mention Mississippi State University.

Lines 157-159: The study included various patterns of alcohol consumption, including those who do not consume alcohol. Alcohol consumption was not an inclusion criterion, and the AUDIT-C does not account for drinking alcohol once a month. A range of alcohol intake with mean and standard deviation is provided.

Conclusions:

A suggestion has been included for future studies to incorporate an actual taste test to examine correlations between TAS2R and the ability to perceive taste quality.

Reviewer 2 Report

Comments and Suggestions for Authors

The topic of this study is an interesting paper on alcohol consumption, body fat, and taste genes.

However, overall, many parts only describe the results without presenting a table of results, disrupting the overall flow of the paper. Therefore, I think, it is necessary to add the tables pointed out below and modify the results.

p88~91

1.     The title of this study is for alcohol consumption, body fat, and taste genes. However, the research method for alcohol consumption is explained too simply. More explanations are needed:  the types of alcohols investigated, how to calculate alcohol intake, etc.

p145~146

2.     Please include more variables in Table 1.: such as body size (height, weight), smoking status, alcohol intake, and body composition (free fat mass, body fat mass, obesity level, etc.).

p150~151

3.     Please delete: df = C-1 = 3-1

4.     Please delete the variable columns after the p-value (European American population African American population, Allele frequency) in Table 2

p153~165

5.     Please revise the subtitle; the content and subtitle title do not match (contents included only AUDIT questions and information on alcohol intake)

p 166~168

6.     Provide specific alcohol intake (g/d) for both groups.

7.     Also, provide the percentage of energy resulting from alcohol consumption (% Kcal/d).

p184

8. There are so many unnecessary parts in Table 3.

â‘       Keep only variables (genotype, p-value, by age group, total, like TT-CT: -2.077, TT-CC: -2.315, CT-CC: -0.510)

â‘¡      -Statistical methods (like Pairwise test comparison Bonferroni correction) and minimum significance levels) should indicate as comments below the table.

â‘¢      Delete “H(2) and effect size within table 3”

â‘£      But please Add the average values for the blowing variables in table 3.

⑤      average scores of Q1, Q2, Q3 by group (ALL, AGE>21, AGE<21)

â‘¥      Averages for Alcohol consumption per gram and Energy from Alcohol (gram) by group (ALL, AGE>21 AGE<21)

P 186~218

9. It is confusing because only the results are described without any tables. Please present additional tables corresponding to the contents (P186~218)

-Delete all PATIAL n2 from the contents

p214~216

10. Is this all content of the results for TAS2R38 (rs1726866, rs10246939, and rs713598)? If so, it is not persuasive to say that “TAS2R38 was studied with just this one sentence (without presenting any result tables including allele frequencies or significances).

â‘       Please either remove all the contents regarding this or add more tables or informations related to the results of TAS2R38.

â‘¡      The sentences (p275~277) in the discussion below also need to be removed or add more result tables for the same reasons as the above. p275~277 “Also, this study could not support the findings presented by Bouthoorn et al. (2014) on the association between TAS2R38 (rs713598) and body fat percentage. The other SNPs 276 (rs1726866 and rs10246939) were also not significant.”

Author Response

Thank you for your detailed feedback on our manuscript. We have addressed your comments as follows:

Comments:

Explanation of Alcohol Consumption:

Concern: The research method for alcohol consumption is explained too simply.

Response: Additional explanations regarding the types of alcohol investigated and the calculation of alcohol intake have been added. Please see lines 115-117.

Variables in Table 1:

Concern: Include more variables in Table 1 such as body size (height, weight), smoking status, alcohol intake, and body composition.

Response: Table 1 has been updated to include these variables (see lines 160-162). Smoking status and waist-hip ratio are still missing.

Line 150-151:

Concern: Delete the formula "df = C-1 = 3-1".

Response: The formula has been removed (line 166).

Columns in Table 2:

Concern: Delete the variable columns after the p-value (European American population, African American population, Allele frequency).

Response: These columns have been deleted (see line 168).

Subtitle Revision:

Concern: Revise the subtitle to match the content.

Response: The subtitle has been corrected (line 169).

Specific Alcohol Intake:

Concern: Provide specific alcohol intake (g/d) for both groups.

Response: Specific alcohol intake values have been added (lines 183-186).

Energy from Alcohol Consumption:

Concern: Provide the percentage of energy resulting from alcohol consumption (% Kcal/d).

Response: This information has been added (lines 183-186).

Table 3 Revisions:

Concern: Unnecessary parts in Table 3.

Keep only relevant variables (genotype, p-value, by age group, total).

Response: Only relevant variables have been retained, and the effect size and Kruskal-Wallis test have been removed.

Statistical methods should be indicated as comments below the table.

Response: Statistical methods are now indicated as comments below the table.

Delete “H(2) and effect size within Table 3”.

Response: These have been deleted.

Add average values for the variables in Table 3.

Response: Average values have been added, including average scores of Q1, Q2, Q3 by group (ALL, AGE>21, AGE<21) with standard error of the mean.

Average scores of Q1, Q2, Q3 by group (ALL, AGE>21, AGE<21).

Response: Text includes averages with standard error of the mean.

Averages for alcohol consumption per gram and energy from alcohol (gram) by group (ALL, AGE>21, AGE<21).

Response: Rank mean has been added to the two age groups.

Results Section Tables:

Concern: Results are described without corresponding tables, which is confusing.

Response: MDPI limits the number of tables to 2-3. Additional tables corresponding to the contents from pages 186-218 have been provided. Please see the text for these tables.

TAS2R38 Results:

Concern: Limited content on TAS2R38 (rs1726866, rs10246939, and rs713598).

Either remove content regarding TAS2R38 or add more tables/information.

Response: A small table (Table 3.b) showing p-values for TAS2R38 has been added. The focus remains on allelic distribution for TAS2R13, which showed some level of significance.

Remove or expand sentences in the discussion regarding TAS2R38.

Response: The sentences on lines 275-277 in the discussion have been revised to align with the updated content (see lines 302-304).

Thank you again for your valuable feedback, which has helped improve the manuscript.

Reviewer 3 Report

Comments and Suggestions for Authors

This manuscript was entitled as “The Influence of Taste Genes on Body Fat and Alcohol Consumption”. The authors concluded that Body fat percentage and alcohol consumption may be influenced by ethnicity, gender and age rather than SNPs of TAS2R13 and TAS2R38 genes.

.          

There are several concerns about this study.

1.         The authors assumed the polymorphisms of taste genes might influence body fat and alcohol consumption. However, in this study, the authors only tested TAS2R13 and TAS2R38. TAS2R38 is one of 25 functional TAS2R genes coding bitter taste receptor proteins, and the authors did not examine the polymorphisms of sweet taste genes (TAS2R genes). Therefore, based on the methodology of this study, it is difficult to make a conclusion that whether taste genes influence body fat and alcohol consumption.

2.         In the section of Results, the data were not presented well. Table 2 shows TAS2R13 (rs1015443) allele frequencies, but TAS2R38 (rs1726866, rs10246939 and rs713598) allele frequencies compared to the American population were not presented.

3.         In the section of Discussion, the authors might discuss the possible influence of other bitter taste genes and sweet genes on body fat and alcohol consumption.

Author Response

Dear Reviewer,

Thank you very much for your insightful comments and suggestions. We have addressed the concerns as follows:

Title and Conclusion:

The manuscript entitled "The Influence of Taste Genes on Body Fat and Alcohol Consumption" concludes that body fat percentage and alcohol consumption may be influenced more by ethnicity, gender, and age rather than by SNPs of TAS2R13 and TAS2R38 genes.

Concerns:

Scope of Taste Genes:

Concern: The study assumed that polymorphisms of taste genes might influence body fat and alcohol consumption, but only tested TAS2R13 and TAS2R38. TAS2R38 is one of the 25 functional TAS2R genes, and the study did not examine sweet taste genes (TAS2R genes).

Response: A paragraph has been added to the introduction (lines 66-70) to address this concern. Additionally, lines 87-89 in the methods and lines 234-237 and 315-316 in the discussion and conclusion sections have been updated to reflect the limitations and rationale behind focusing on these specific genes.

Presentation of Data:

Concern: In the Results section, TAS2R13 (rs1015443) allele frequencies are presented, but TAS2R38 (rs1726866, rs10246939, and rs713598) allele frequencies compared to the American population were not.

Response: Since TAS2R13 showed some influences, the research team decided to focus on its allelic distribution. This decision has been clarified in the Results section to explain why the focus was on TAS2R13.

Discussion on Other Taste Genes:

Concern: The Discussion section should address the possible influence of other bitter taste genes and sweet genes on body fat and alcohol consumption.

Response: This concern has been addressed through updates in the introduction and methods sections, explaining the choice of these SNPs. The discussion section has been enhanced to acknowledge the potential influences of other bitter taste genes and sweet genes on the study outcomes.

Thank you again for your valuable feedback, which has helped improve the manuscript.

Round 2

Reviewer 3 Report

Comments and Suggestions for Authors

This manuscript was entitled as “The Influence of Taste Genes on Body Fat and Alcohol Consumption”. The authors concluded that Body fat percentage and alcohol consumption may be influenced by ethnicity, gender and age rather than SNPs of TAS2R13 and TAS2R38 genes.

.          

There is one suggestion about this study.

1.         In the section of Results, the data were not presented well. Table 2 shows TAS2R13 (rs1015443) allele frequencies, but TAS2R38 (rs1726866, rs10246939 and rs713598) allele frequencies compared to the American population were not presented.

Author Response

Dear reviewer,

Thank you for your time. We have calculated the allele frequencies for the other SNPs.